# Relapsed Ovarian Cancer Patients with Ascites and/or Pleural Effusion Still Benefit from Treatment: A Real-Life Study

**DOI:** 10.3390/cancers16010162

**Published:** 2023-12-28

**Authors:** Mariana Rebordão-Pires, Marta F. Estrada, António Gomes, Filipa Silva, Carlota Baptista, Maria João Ramos, Ana Fortuna, Pedro Simões, Gabriela Sousa, Ana Marreiros, Rita Fior

**Affiliations:** 1Medical Oncology Unit, Instituto Português De Oncologia De Coimbra Francisco Gentil, 3000-075 Coimbra, Portugal; mariana.pires@chln.min-saude.pt (M.R.-P.); 3036@ipocoimbra.min-saude.pt (G.S.); 2Cancer Development and Innate Immune Evasion Laboratory, Champalimaud Foundation, 1400-038 Lisboa, Portugal; 3Surgery Unit, Hospital Vila Franca de Xira, 2600-153 Vila Franca de Xira, Portugal; antonio.gomes2@hvfx.min-saude.pt; 4Gynecology Unit, Champalimaud Foundation, 1400-038 Lisboa, Portugal; filipa.silva@fundacaochampalimaud.pt; 5Medical Oncology Unit, Hospital Beatriz Ângelo, 2674-514 Loures, Portugal; carlota.vieira.baptista@hbeatrizangelo.pt (C.B.); pedro.simoes@hbeatrizangelo.pt (P.S.); 6Medical Oncology Unit, Centro Hospitalar Universitário de Santo António, 4099-001 Porto, Portugal; u12994@chporto.min-saude.pt; 7Medical Oncology Unit, Centro Hospitalar do Algarve, 8500-338 Portimão, Portugal; afortuna@chalgarve.min-saude.pt; 8Faculty of Medicine and Biomedical Sciences, University of Algarve, 8005-139 Faro, Portugal; ammarreiros@ualg.pt; 9Algarve Biomedical Center Research Institute, University of Algarve, 8005-139 Faro, Portugal

**Keywords:** relapsed ovarian cancer, ascites, pleural effusion, overall survival

## Abstract

**Simple Summary:**

The majority of high grade serous ovarian carcinoma (HGSOC) patients present with advanced disease, which will relapse and require additional treatment. Although the presence of ascites and/or pleural effusion in relapsed HGSOC is not rare, there is a lack of data in this setting. This population of patients typically presents a very poor prognosis and often there is no response to the treatment. One could ask if giving the “right therapy to the right patients” would change the course of the disease. The aim of this retrospective, multi-centric and real-life study was to compare the overall survival (OS) of two groups of advanced HGSOC patients, who present with ascites or pleural effusion: responders, i.e., patients that had an imagological response to treatment (complete/partial response/stable disease, RECIST criteria) versus non-responders (no response/progression upon treatment), and to evaluate the predictive value of the standard clinical variables. We concluded that patients who have a tumoral response to first and second line of treatment have a ~3 times longer Progression-free Survival (PFS) or OS. The response to the treatment was irrespective of patient and tumor characteristics, which highlights the need for a sensitivity test to tailor treatments and improve efficacy rates in a personalized manner.

**Abstract:**

(1) Background: Relapsed HGSOC with ascites and/or pleural effusion is a poor-prognostic population and poorly represented in clinical studies. We questioned if these patients are worth treating. In other words, if these patients received the most effective treatment, would it change the course of this disease? To our knowledge this is the first real-life study to evaluate this question in this low-survival population. (2) Methods: To tackle this question we performed a retrospective, multi-centric, real-life study, that reviewed relapsed HGSOC patients with ascites and/or pleural effusion. Our rationale was to compare the OS of two groups of patients: responders, i.e., patients who had an imagological response to treatment (complete/partial response/stable disease, RECIST criteria) versus non-responders (no response/progression upon treatment). We evaluated the predictive value of clinical variables that are available in a real-life setting (e.g., staging, chemotherapy, surgery, platinum-sensitivity). Multivariate logistic regression and survival analysis was conducted. A two-step cluster analysis SPSS tool was used for subgroup analysis. Platinum sensitivity/resistance was also analyzed, as well as multivariate and cluster analysis. (3) Results: We included 57 patients, 41.4% first line responders and 59.6% non-responders. The median OS of responders was 23 months versus 8 months in non-responders (*p* < 0.001). This difference was verified in platinum-sensitive (mOS 28 months vs. 8 months, *p* < 0.001) and platinum-resistant populations (mOS 16 months vs. 7 months, *p* < 0.001). Thirty-one patients reached the second line, of which only 10.3% responded to treatment. Three patients out of thirty-one who did not respond in the first line of relapse, responded in the second line. In the second line, the mOS for the responders’ group vs. non-responders was 31 months versus 13 months (*p* = 0.02). The two step cluster analysis tool found two different subgroups with different prognoses based on overall response rate, according to consolidation chemotherapy, neoadjuvant chemotherapy, FIGO staging and surgical treatment. Cluster analysis showed that even patients with standard clinical and treatment variables associated with poor prognosis might achieve treatment response (the opposite being also true). (4) Conclusions: Our data clearly show that relapsed HGSOC patients benefit from treatment. If given an effective treatment upfront, this can lead to a ~3 times increase in mOS for these patients. Moreover, this was irrespective of patient disease and treatment characteristics. Our results highlight the urgent need for a sensitivity test to tailor treatments and improve efficacy rates in a personalized manner.

## 1. Introduction

Ovarian cancer is the third most common type of gynecologic cancer, after cervical and uterine cancer. Disease signs and symptoms are often non-specific, leading to late diagnosis. As such, the majority of patients presents with advanced disease stages (III or IV), have a poor prognosis and a 5-year survival of less than 30% [1]. Initial therapy usually consists of surgical cytoreduction, followed by platinum-taxane combination therapy and maintenance therapy with anti-angiogenic agents (the monoclonal antibody, bevacizumab) or poly (ADP-ribose) polymerase (PARP) inhibitors (niraparib and olaparib) [1]. However, neoadjuvant chemotherapy (NACT) prior to definitive surgery is an alternative option in selected patients: those with advanced ovarian cancer (stage III and IV) with clinically apparent, unresectable disease who are unlikely to become complete or optimally cytoreduced; or those with ovarian cancer at any stage who are poor operative candidates, but who are likely to tolerate surgery after NACT. Typically, three to four cycles of NACT are administered, after which resectability is assessed by imaging. For those who are candidates, interval cytoreduction is performed, followed subsequently by further cycles of chemotherapy in the adjuvant setting. For those who do not have FF disease after the initial cycles of NACT, primary medical therapy is offered, without surgery. The goal of NACT is to reduce perioperative morbidity and mortality and increase the likelihood of a complete resection of disease at the time of cytoreductive surgery. Despite the potentially improved surgical outcomes that result from NACT, clinical studies have shown that survival is not improved with NACT followed by surgery versus standard surgery followed by adjuvant chemotherapy [2,3,4].

Despite the advanced stage, ~80% of women present with a good initial response [5]. Nonetheless, the majority will relapse and require additional treatment [6], within the first two years [7]. The management of relapsed disease is defined based on the time that has elapsed between the completion of platinum-based treatment and the detection of relapse, known as the platinum-free interval (PFI). The definition of PFI has been specified at the Fourth Vancouver Ovarian Cancer Consensus Conference in 2010 [8]. This is due to the correlation between PFI with progression-free survival (PFS), overall survival (OS), and response to subsequent treatment (both with platinum and non-platinum agents, as well as cytoreduction). More specifically:-Patients with a PFI of six months or longer are “platinum-sensitive” [9].-Patients with a PFI of less than six months are considered platinum-resistant. Of this group, those patients who progress while on platinum-based therapy are often referred to as having “platinum-refractory” disease [6].

According to PFI, there are different treatment options according to each setting. For platinum-sensitive disease, the preferable regimens according to NCCN guidelines are carboplatin/paclitaxel, carboplatin/gemcitabine, and carboplatin/liposomal doxorubicin, all of these with or without bevacizumab. For platinum-resistant disease, the potential options include paclitaxel, docetaxel, etoposide, liposomal doxorubicin, gemcitabine, topotecan, and bevacizumab. Choice among these agents depends upon the clinician’s experience, patient’s comorbidities, the side effect profile, and prior therapy [10].

In advanced stage patients, there is a frequent accumulation of ascites (fluid in the abdomen). This fluid is composed of both acellular and cellular components, such as tumor cells, stromal cells, inflammatory cells and endothelial cells [11]. This fluid is known to contribute to patient morbidity and mortality by facilitating metastasis and contributing to chemo-resistance [11]. Malignant ascites promote tumor dissemination to intra-peritoneal sites by means of biochemical and physical cues which have been modeled experimentally using ex vivo analysis of human samples and in vitro methods with cell lines [7,12]. In order to relieve some patient’s symptoms (for example, shortness of breath or obstipation), a paracentesis is usually performed to remove the fluid.

The presence of pleural effusion in the evolution of patients with ovarian cancer is also common. In a series of 123 patients, malignant pleural effusion (MPE) at diagnosis was observed in 29%, and in 70% during the course of the disease. In one study, patients with ovarian cancer and pleural effusion, at diagnosis or during the course of the disease, experienced reduced survival compared to patients without MPE [13]. Occasionally, it is indicated to perform a thoracentesis to relieve patient’s symptoms related to pleural effusion.

Despite the worse prognosis of ovarian cancer relapsed patients with accumulation of ascites and/or pleural effusions, there are no real-life studies focusing on this population. How do these patients respond to therapy? Is there a clinical benefit in treating these patients? Is it possible to observe differences between responders and non-responders? One could ask if this population, with very poor prognosis, would still benefit from treatment, and if given the right therapy, could this change the course of the disease and the OS? Our primary goal was to compare the OS (primary endpoint) of responders and non-responder patients, according to the imagological response to therapy.

## 2. Materials and Methods

Medical files were retrospectively reviewed from patients with relapsed HGSOC with ascites and/or MPE, from five Portuguese cancer centers. Patients were diagnosed with HGSOC between 2016 and October 2021 and were required to have a minimum follow-up of 12 months.

### 2.1. Data Collection

Electronic medical records were queried for the following variables: age at diagnosis, CA125 at diagnosis, date of diagnosis, use of neoadjuvant chemotherapy, surgery and date of surgery, consolidation chemotherapy, iPARP or bevacizumab maintenance, date of last platinum cycle, date of relapse, type of fluid (ascitic and/or pleural fluid), type of chemotherapy 1st line after relapse, number of cycles 1st line, best response 1st line (according to RECIST criteria: complete/partial response, stable disease, progressive disease), date of progression, type of chemotherapy 2nd line after relapse, number of cycles 2nd line, best response 2nd line (according to RECIST criteria: complete/partial response, stable disease, progressive disease), date of progression, type of chemotherapy 3rd line after relapse, number of cycles 3rd line, best response 3rd line (according to RECIST criteria: complete/partial response, stable disease, progressive disease), date of last follow-up/death.

Imagological tumoral response was evaluated according to RECIST criteria by radiologist experts:Complete response: no visible tumor lesions on computed tomography (CT) or Magnetic Resonance Imaging (MRI) scansPartial response: reduction of at least 30% in target tumor lesionsStable disease: neither sufficient shrinkage for partial response nor an increase in lesions, which would qualify as progressive diseaseProgression: at least 20% increase in the sum of the diameter of target tumor lesions or de novo metastatic sites

Patients were classified as responders if they had a complete/partial response or stable disease, and as non-responders if they had progression, according to RECIST criteria. The best response for each treatment in each line was considered.

### 2.2. Sample Size

The sample size was calculated to compare OS of relapsed HGSOC patients with ascites and/or pleural effusion, between responder and non-responder groups. Assuming a hazard ratio in favor of responders of 0.3 for OS, a total of 27 per group (according to the UCSF Clinical and Translation Science Institute) OS events would be needed for the study to have at least 80% power at a two-sided α level of 0.05 to detect a statistically significant effect using a survival analysis test for OS, in two well balanced groups: responders and non-responders group.

### 2.3. Statistical Analysis

Disease control rate was defined as the percentage of patients who achieved complete response, partial response and stable disease, and thus the percentage of responders.

Progression free survival was measured as the time from initiation of treatment to the occurrence of disease progression. According to the line of therapy, it will be defined as PFS1 (1st line of therapy) and PFS2 (2nd line of therapy).

OS was measured as the time elapsed from disease relapse to patient death or last follow-up.

According to sample size, descriptive statistics and univariate non-parametric analysis were used.

The primary objective was the comparison of OS between responders and non-responders in each line of treatment.

The software SPSS^®^ V27 was used for all analyses. Cluster analysis was performed with the SPSS “two step cluster analysis” tool. The Kaplan–Meier method and Cox proportional hazard regression model were used for calculation of survival over time, while survival differences were analyzed using the log-rank test. Multivariate analysis was performed with logistic regression and Cox regression models, with ORR and OS as the independent variables, respectively. Variables with statistical significance in the univariate or variables with biological plausibility were included in the multivariate models for a maximum of 4 independent variables per model, according to sample size.

## 3. Results

### 3.1. Study Population

In order to investigate how HGSOC relapsed patients with accumulation of ascites and/or pleural effusions respond to therapy and whether there is a clinical benefit of treatment, we studied retrospectively a cohort of 57 patients from five cancer centers.

Median age at diagnosis was 67 years old (48–83). The median follow-up time was 48 months (16–71). ECOG performance status, CA125, histological types, FIGO staging at diagnosis, therapeutic approaches and the type of fluid present at relapse were analyzed and are presented in Table 1.

At relapse, 43.9% of the patients were platinum-sensitive (n = 25) and 56.1% were platinum-resistant (n = 32). In addition, 40.4% experienced platinum combo at first relapse (n = 23), two patients platinum- (3.5%), and 56.1% (n = 32) therapy without platinum. Most platinum-resistant patients received Bevacizumab treatment, but we did not collect this data.

In the first line of relapse, 24 patients were categorized as responders: 1 patient had complete response, 18 patients had partial response and 5 patients had stable disease. The disease control rate was 42.1%.

### 3.2. Natural Dataset Grouping—Cluster Analysis Defines Three Clusters

After gathering all variables for the population, we started by performing a two step cluster analysis with SPSS software V27. Our goal was to identify which groups had a better response to treatment and if there were variables/conditions that could predict response and therefore could bias our OS and PFS analysis.

The two step cluster analysis revealed three different clusters based on four different variables: neoadjuvant chemotherapy (NACT), consolidation chemotherapy (CT), surgical treatment and FIGO staging (Figure 1a).

These three clusters have different prognostic characteristics with different ORR, OS and PFS (Figure 1a).

Cluster 1 (n = 17) includes patients who were surgically resected, not submitted to neoadjuvant chemotherapy, and all, except one, had been submitted to consolidation chemotherapy. This cluster had the higher proportion of ORR, higher OS and PFS (Figure 1a—yellow).

In contrast, in Cluster 3 (n = 17), the majority of patients were stage IV, were submitted to neoadjuvant chemotherapy, more than half were not resected, and none were submitted to consolidation chemotherapy. This was the cluster with worse prognosis (Figure 1a—blue).

Twenty-two patients belonged to an in-between prognostic cluster. Patients were stage III or IV, all were submitted to surgical resection and consolidation chemotherapy, and most were submitted to neoadjuvant chemotherapy (Figure 1a—red).

Figure 1b,c shows the relative importance of each variable for cluster model definition and the silhouette measure of cohesion and separation, with an average of 0.6 being considered a good quality model.

These data reveal, as expected, that early Stage FIGO III patients who went through surgical resection with no NACT, but who performed consolidation CT, had a better prognosis with higher ORR. However, within this cluster and the other clusters, there were always some patients who responded to therapy and others that did not.

Moreover, if we analyze more closely the predictor variables of cluster 1, one by one (consolidation therapy; no NACT; surgery and Stage FIGO III) and plot the OS of responders and non-responders in each category, there is always a statistically significant difference between the two groups (Figure 2).

These results suggest that, even if there are predictors of response, i.e., that indicate a better prognosis for one group, there will always be some patients who will respond better to treatment than others within that group.

### 3.3. PFS and OS of Responder-Patients Is Significantly Longer than Non-Responder Patients at First Line of Relapse

Having discarded any confounding variables that could bias our analysis, we first started by analyzing the PFS and OS at first line of relapse in responders and non-responders. Median PFS in the first line of relapse (PFS1) in the study population was 4 months (2.5–5.5). In the responder’s group (n = 25, 43.8%), the median PFS1 was 11 months versus 3 months (*p* < 0.0001) in the non-responder’s group (n = 32, 56.1%), as illustrated in Figure 3a. Regarding OS, in the responder’s group, the median OS was 23 months (15.2–30.7) versus 8 months (6.2–9.8) in the non-responder’s group (*p* < 0.001), (Figure 3b).

Next, we questioned whether this same trend applies for platinum sensitive and resistant patients. Our results show that the differences in PFS and OS were independent of the platinum sensibility status. In the platinum-sensitive group, the median PFS1 was 12 months (4.9–19) in the responder’s group versus 4 months (1.1–6.9) in the non-responder’s group (*p* < 0.0001) (Figure 3c). In the platinum-resistant group, the median PFS1 was 8 months (2.8–13.1) in the responder’s group versus 3 months (1.9–4.1) in the non-responder’s group (*p* < 0.0001) (Figure 3d).

Regarding OS, in the platinum-sensitive group, mOS was 28 months (22.7–33.5) in responders vs. 8 months in non-responders [5.1–11], CI 95, *p* < 0.0001) (Figure 3e). In the platinum-resistant population, mOS was 16 months [6.3–25.7] in responders vs. 7 months in (3.6–10.4) non-responders, CI 95, *p* < 0.001) (Figure 3f). As expected, the median OS for platinum-sensitive patients was higher than for platinum-resistant patients—19 months (8.6–29.4) versus 9 months (5.0–13.0), but not statistically significant (*p* = 0.072) (Figure 4).

### 3.4. PFS and OS at Second Line of Relapse in Responder-Patients Is Significantly Longer than Non-Responders

From 57 patients, only 30 patients reached second line of relapse (52.6%). From these, 6 patients were classified as responders. The disease control rate was 10.5%. In the responder’s group, the median progression-free survival was 6 months (2.8–9.2) versus 2 months (1.4–2.7) in the responder’s group (*p* = 0.008) (Figure 5a).

In relation to the median OS, the responder’s group had a mOS 2.4 times longer than non-responders: 31 months (10.3–15.7) versus 13 months (10.3–15.7) (*p* = 0.02) (Figure 5b). At the end of the study, 44 patients had died (77.2%). The study population median OS at the moment of relapse was 12 months (8.0–15.9).

In the platinum-sensitive group, the median PFS2 was 4 months in the responders’ group versus 1 month in the non-responders’ group (*p* = 0.003). In the platinum-resistant group, the median PFS2 was 6 months in the responders’ group versus 2 months in the non-responders’ group (*p* = 0.003), as can be observed in Figure 5c,d.

Next, we analyzed the correlation between disease control rate (percentage of patients who achieved a complete/partial response or stable disease) and OS. Our data indicate that disease control rate was strongly correlated with OS in the first line of relapse (Pearson Coefficient 0.5, *p* < 0.0001) and moderately correlated in the second line of relapse (Pearson Coefficient 0.45, *p* = 0.014).

Binary regression logistic models were tested considering responders/non-responders for a maximum of four independent variables per model, according to sample size. CA125 at diagnosis, neoadjuvant chemotherapy, Primary Surgery, FIGO staging, and consolidation chemotherapy were not independent predictors of responders/non-responders.

### 3.5. Sequence of Response

Finally, we analyzed the sequence of response in first and second line of relapse. Our goal was to investigate if a non-responder (NR) in 1st line will always be a NR, or whether the patient can then respond to 2nd line, and the same question was asked for 1st line responders (R).

Our results show that most patients (80%) do not respond to 2nd lines of treatment, irrespective of the 1st line response (Table 2). However, we found exceptions, where 1st line NR become responders (NR^1^ >> R^2^) and 1st line responders still respond to 2nd lines of treatment (R^1^ >> R^2^). Also, as expected, patients that respond to both lines of treatment have longer mOS.

## 4. Discussion

Relapsed HGSOC with ascites and/or pleural effusion is associated with poor survival [12]. As such, we questioned if therapy is able to change the course of the disease.

In the published literature, this question remains unanswered, as published data with these patients, with or without fluid accumulation, are mostly based on highly controlled clinical trials. To address this, we performed a retrospective, multi-centric, real-life study with relapsed HGSOC patients who present with ascites and/or pleural effusion. We categorized patients as responders if they had an imagological complete/partial or stable disease, according to RECIST criteria. The patients who progressed were categorized as non-responders. We compared survival results in both groups, in first and second line of relapse.

In the first line of relapse, strikingly we observed ~3 times longer OS in responders (mOS 23 m vs. 8 m). We identified responders and non-responders in the platinum-sensitive population (mOS 28 months vs. 8 months, *p* < 0.001) and in the platinum-resistant population (mOS 16 months vs. 7 months, *p* < 0.001). The median OS was higher in the platinum-sensitive population (mOS 19 m vs. 9 m, *p* = 0.07), which is similar to the literature reports [14,15,16]. Nevertheless, the most significant difference in OS was between responders and non-responders.

Nowadays, in clinical practice, the treatment decisions in HGSOC relapse, are based on platinum-free interval, according to ESMO and NCCN guidelines. These guidelines stratify patients as platinum-sensitive (PFI of six months or longer), and platinum-resistant (PFI of less than six months) [9]. In each setting of sensitivity or resistance, there are numerous treatment options that clinicians should discuss with the patient. Usually the treatment is decided in partnership with the patient, based on its toxicity profile and patient comorbidities, and very often there is not a “right answer” [17]. As such, patients often go through rounds of trial and error to find the best therapy option that controls the disease.

In our real-life study, we asked whether patient stratification according to the platinum-sensitivity status itself could be correlated with OS. Our results show, that although there is a longer mOS in platinum-sensitive patients, we did not verify a statistically significant difference. In contrast, previous studies have shown that platinum-free interval correlates with survival results [14]. However, our study was focused only on HGSOC patients with ascites and/or pleural effusion, which is already a poorer prognosis group of patients [7,11,12,13], which may explain this difference.

In this study, there was a strong and positive correlation between disease control rate and OS. This means that being a responder (complete/partial response or stable disease) can predict a higher OS.

In order to predict patient response to therapy to tailor treatments, different research groups have developed different approaches, like mouse and zebrafish patient derived xenografts, or Avatars [18,19,20], or ex vivo platforms using tumor fragments/organoids [21,22]. The main goal of these predictive personalized tests is to transform non-responders into responders. These types of test could also help oncologists decide if a patient will benefit from treatment or if there is no benefit in exposing a specific patient to ineffective therapies that do not reduce tumor burden but will always lead to high toxicities and side-effects.

Another important point to discuss is the median OS. Usually, in the most important phase III trials, the median OS for relapsed disease is 24–29 months for platinum-sensitive disease [14] and 11–14 months for platinum-resistant disease [23]. In our population, the median OS is 12 months, 19 months in the platinum-sensitive population and 9 months in the platinum-resistant population. These discrepancies might be explained by our specific cohort with poorer prognosis (cancer patients with ascites and/or pleural effusion), and not the total population of HGSOC patients.

Almost 50% of tumors did not respond to any line of treatment (n = 14, mOS 13.4 m) or, when the tumor responded in first line, it did not respond in the second line of relapse (n = 10, mOS 21.2 m). However, there were three patients who obtained a response in the second line of relapse, when the tumor did not respond in the first line. The median OS of those patients was similar to that for having a response in the first line (mOS 21.3 m). It is imperative to make efforts to transform all patients into responders in first and second line of the relapse settings, which could prolong by more than 2 years the median OS, according to our study (n = 3, mOS 33 m).

In the literature, the ICON4/OVAR2.2 study compared single-agent carboplatin with a doublet regime of carboplatin and paclitaxel. This study demonstrated a 7% improvement in 2-year survival rate in the combination arm (57% versus 50% in the monotherapy arm). It also showed a 5-month improvement in median survival (29 versus 24 months) [14].The Arbeitsgemeinschaft Gynaekologische Onkologie (AGO) group demonstrated the role of doublet carboplatin-gemcitabine chemotherapy in platinum sensitive recurrent ovarian cancer. In this study, patients received a median of six cycles of chemotherapy in both arms improving the median PFS (8.6 versus 5.8 months). Despite these improvements in PFS, there were no significant differences in quality of life scores [15]. In platinum-resistant disease, a Cochrane review assessing the efficacy and safety of pegylated liposomal doxorubicin (PLD) in relapsed epithelial ovarian cancer found that the median PFS was ~3 months and the OS for PLD monotherapy was ~13 months [24]. A meta-analysis of six randomized controlled trials compared gemcitabine to PLD in the setting of platinum-resistant recurrent ovarian cancer, and demonstrated no significant differences in OS or PFS [25]. We did not find any published studies that evaluate patient survival according to the sequence of response in different lines of treatment.

PFS represents the time of survival for each treatment line. In both lines of relapse, median PFS was ~3 times higher in patients who had a favorable imagological response. OS was also significantly longer in responder patients. These data suggest that it is definitely worth treating patients with the “right therapy”, in order to achieve the best clinical response. However, many patients are multi-resistant and therefore it is impossible to find the best treatment option in the guidelines. However, with a personalized screening test, these multi-resistant patients could be identified and given the possibility of testing new off-label options.

These results lead us to conclude that, even when isolating patients with similar disease characteristics, either related to better or worse prognosis, there will always be patients that respond and patients that do not respond to treatment. In other words, it is not possible to predict the tumor response to treatment based on the clinical characteristics, namely the stage of tumor progression, or subjection to neoadjuvant chemotherapy, surgery, or consolidation chemotherapy. We hypothesize that a sensitivity test could help optimize treatment, identifying the “right treatment” for the “right patient”, and eventually transform non-responders into responders and thus improve the prognosis of these patients.

The conclusions taken from this study need to consider the small sample size of 57 patients. Logistic regression is very dependent on sample size and the results should be taken cautiously bearing in mind that type II error is not low enough to establish definitive conclusions regarding “absence” of independent risk factors.

In addition, as a retrospective study it presents some disadvantages, namely regarding data collection, but new hypotheses have also been created to be then tested in prospective studies.

Despite these limitations, this is the first survival multi-centric study performed in relapsed ovarian cancer patients, with malignant ascites and/or pleural effusion.

## 5. Conclusions

In our retrospective, multi-centric, real-life study, we observed a ~3-fold increase in OS, PFS1 and PFS2 in the group of patients who had a favorable imagological tumoral response (complete/partial response and stable disease). Cluster analysis also led us to conclude that, even when isolating patients with similar disease characteristics, either related to better or worse prognosis, we will always find patients that respond and patients that do not respond to treatment. Why do some relapsed HGSOC with ascites and/or pleural effusion respond to treatment and others do not is a complex question and probably due to the tumor-specific sensitivity to the treatment choices. A personalized sensitivity test that could anticipate tumoral response could be key in treating these patients and prolonging their OS.

Importantly, our data clearly show that these patients will benefit from treatment, since there is a subgroup of patients that responds to treatment and thus has a higher OS. The next step should be an interventive clinical trial to understand if a personalized sensitivity test that screens all options beforehand could transform non-responders into responders. Such a test could optimize treatments, improving the PFS and OS of these patients and sparing them from the toxic effects of ineffective treatments.

## Figures and Tables

**Figure 1 cancers-16-00162-f001:**
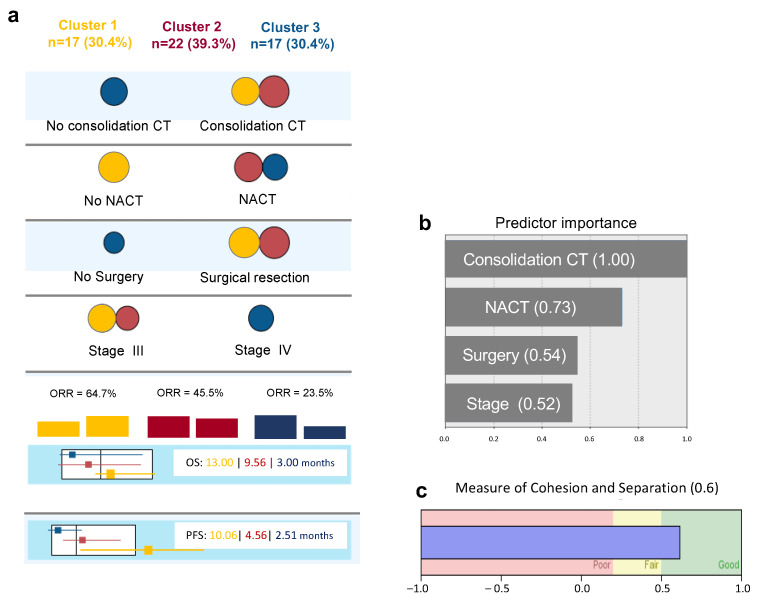
Diagram of Cluster Analysis: (**a**) Distribution of predictive variables according to each cluster (circle’s area is proportional to number of patients of each cluster in each variable—since variables are mutually exclusive, only the most frequent category is represented in each variable for each cluster). (**b**) Relative importance of predictive variables in cluster definition. (**c**) Model quality according to cohesion and separation measurement. CT—Chemotherapy/NACT—Neoadjuvant Chemotherapy/OS—Overall Survival/PFS—Progression Free Survival.

**Figure 2 cancers-16-00162-f002:**
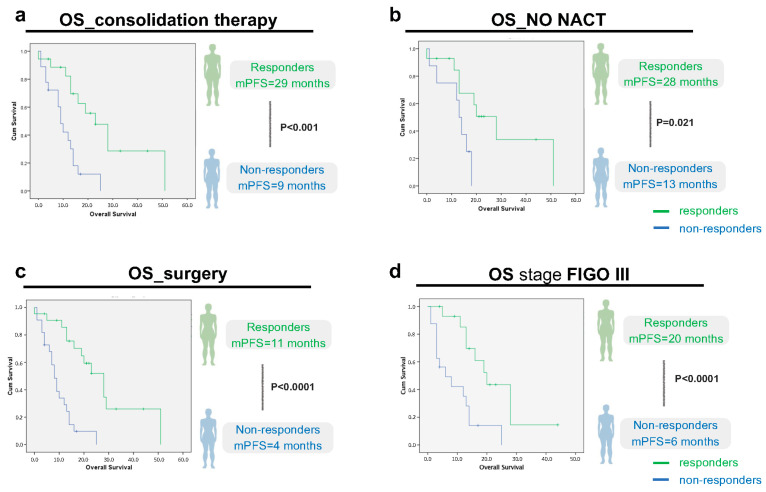
Overall survival (OS) analysis, according to imagological response (responders/non-responders) within the predictor variables of cluster 1: (**a**): consolidation therapy (*p* < 0.0001, CI 95, n = 18 responders vs. 18 non-responders); (**b**): no NACT (*p* = 0.021, CI 95, n = 14 responders vs. 8 non-responders); (**c**): surgery (*p* < 0.0001, CI 95, n = 22 responders vs. 22 non-responders); and (**d**): stage FIGO III (*p* < 0.0001, CI 95, n = 15 responders vs. 16 non-responders). Green line: Responders (Complete/Partial Response or Stable Disease). Blue line: Non-Responders (No response/Progressive disease).

**Figure 3 cancers-16-00162-f003:**
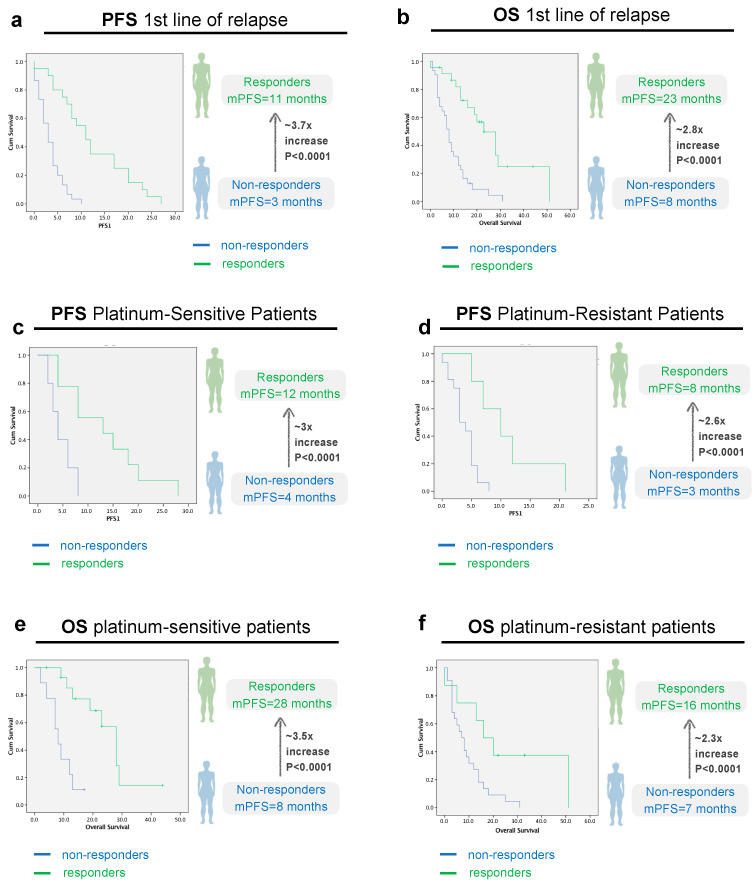
Progression-Free Survival (PFS) and Overall survival (OS) analysis in the first line of relapse, according to imagological response, all patients (**a**,**b**) and according to platinum-sensitivity status (**c**–**f**). (**a**): PFS 1st line of relapse, all patients (*p* < 0.0001, CI 95, n responders = 25 vs. 32 non-responders); (**b**): OS 1st line of relapse, all patients (*p* < 0.0001, CI 95, n responders = 25 vs. 32 non-responders); (**c**): PFS platinum-sensitive patients (*p* < 0.0001, CI 95, n responders = 16 vs. 9 non-responders); (**d**): PFS platinum-resistant patients (*p* < 0.0001, CI 95, n responders = 9 vs. 23 non-responders); Green line: Responders (Complete/Partial Response or Stable Disease). Blue line: Non-Responders (No response/Progressive disease).

**Figure 4 cancers-16-00162-f004:**
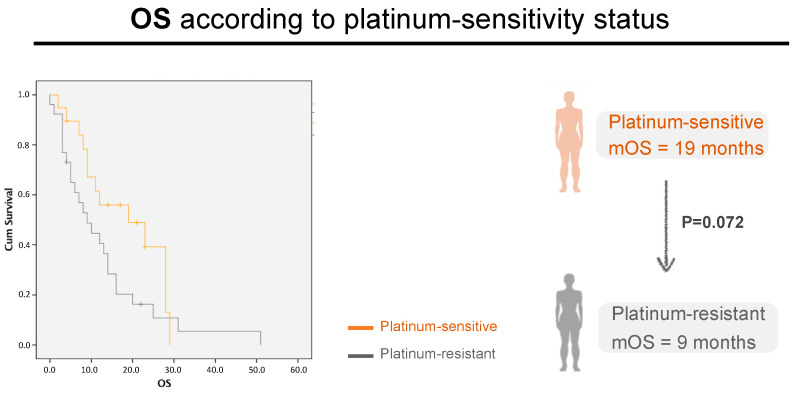
Patient’s OS in the first line of relapse, according to platinum-sensitivity Status. Grey line: platinum-resistant patients. Orange line: platinum-resistant patients; *p* = 0.072, CI 95, n = 25 vs. 32, respectively.

**Figure 5 cancers-16-00162-f005:**
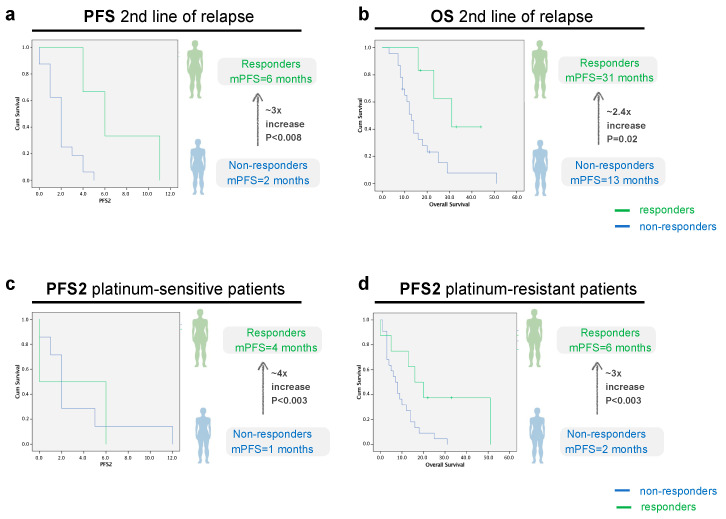
PFS and OS analysis in the second line of relapse, according to imagological response to treatment, all patients ((**a**): *p* = 0.008, CI 95, n = 6 responders vs. 24 non-responders, (**b**): *p* = 0.02, CI 95, n = 6 responders vs. 24 non-responders patients) and sorted according to platinum-sensitivity status ((**c**): *p* = 0.003, CI 95, n = 3 responders vs. n = 11 non-responders, (**d**): *p* = 0.003, CI 95, n = 3 responders vs. n = 13 non-responders). Green line: Responders (Complete/Partial Response or Stable Disease). Blue line: Non-Responders (Progressive Disease).

**Table 1 cancers-16-00162-t001:** Study Population Baseline Characteristics.

Study Population Baseline Characteristics (n = 57)	
Age at diagnosis—years, median (min–max)	67 (48–83)
ECOG/PS at diagnosis, n (%)	
0	11 (19.3%)
1	46 (80.7%)
CA125 at diagnosis—U/L, average (σ)	3156 (14,702)
Histological types, n (%)	
Serous High-Grade, n (%)	57 (100%)
Neoadjuvant Chemotherapy, n (%)	35 (61.4%)
Primary surgery, n (%)	46 (80.7%)
FIGO staging–n (%)	
IC, n (%)	1 (1.8%)
IIIA, n (%)	6 (10.5%)
IIIB, n (%)	4 (7%)
IIIC, n (%)	22 (38.6%)
IVA, n (%)	11 (19.3%)
IVB, n (%)	13 (22.8%)
Consolidation Chemotherapy, n (%)	38 (66.7%)
BRCA mutation, n (%)	5 (8.8%)
iPARP after consolidation CT, n (%)	5 (8.8%)
Bevacizumab after consolidation CT, n (%)	9 (15.8%)
Ascites at relapse, n (%)	42 (73.7%)
Pleural effusion at relapse, n (%)	6 (10.5%)
Ascites and pleural effusion at relapse, n (%)	9 (15.8%)

**Table 2 cancers-16-00162-t002:** Sequence of Tumoral Response between First Line of Relapse and Second Line of Relapse and respective mOS.

Sequence of Response	n	mOS (Months)
NR^1^ >> NR^2^	14	13.4
R^1^ >> NR^2^	10	21.2
NR^1^ >> R^2^	3	21.3
R^1^ >> R^2^	3	33

^1^ first line of relapse; ^2^ second line of relapse; NR: Non-Responder. R: Responder.

## Data Availability

The data are not publicly available due to ethical constraints.

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
