# Peer review of "Relapsed Ovarian Cancer Patients with Ascites and/or Pleural Effusion Still Benefit from Treatment: A Real-Life Study"

_cancers, 2023, doi:10.3390/cancers16010162_

Round 1
Reviewer 1 Report
Comments and Suggestions for Authors
MDPI Cancers – 2764922
Mariana Rebordai-Pires FIRST AUTHOR
Relapsed Ovarian Cancer patients with ascites and/or pleural effusion are still worth to treat: a real-life study
The manuscript by Rebordai-Pires et al evaluates the treatment options for ovarian cancer patients who have relapsed with ascites and/or pleural effusion. In a study of 57 patients, the median overall survival of first-line responders was significantly longer than non-responders and the authors conclude that it is worthwhile to treat relapsed patients with first- and second-line therapies.
The problem this reviewer has with the manuscript is that the data are hardly surprising. It is well known by case reports and studies evaluating the relative effectiveness of different treatments after standard surgery (such as adjuvant chemotherapy, immunotherapy, and radiological intervention) that a small proportion of patients achieve remission for some period of time. Nonetheless, the data provided are a useful addition to the literature. The authors recommend tailoring treatments to improve efficacy.
Overall, the manuscript is well-written; however, there is some awkward English phrasing in the title and in various places in the text – specifically the phrase “patients….are still worth to treat.”
See details below for necessary revisions:
Title: REVISE TITLE “Treatment of ovarian cancer patients with ascites and/or pleural effusion: a real-life study.”
Simple Summary (lines 19-21): “One could ask whether it is worthwhile to treat a poor prognostic population…”
Abstract (line 27): PFS should be spelled out for the first time in the manuscript as Progression-Free Survival
Abstract (lines 31-32): “We questioned whether it is worthwhile to treat these patients.”
Abstract (lines 53-54): “Our data clearly show that it is worthwhile to treat relapsed ovarian cancer patients.”
Introduction (lines 127-128): “Once could ask whether it is worthwhile to treat such a poor prognostic population or whether, if gven the right therapy, treatment could change the course of the disease and the OS.”
Results (line 191): typographical error: whether
Results (line 292): non-responder’s group
Discussion (line 358): verify
Discussion (lines 370-371): “Also, these types of tests cold help oncologists determine if it is worthwhile to treat….”
Discussion (lines 386-387): “which could prolon the median OS to more than 2 years”
Conclusions (line 432): “Importantly, our data clearly show that it is worthwhile to treat these patients…”
na
Reviewer 2 Report
Comments and Suggestions for Authors
The authors have done a multicentric retrospective study on HGSOC patients, who have had ascites and/or pleural effusion as a feature in first relapse, and looked how they have responded to treatment, and if the treatment has been useful. The main conclusion the authors state, is that if the patient responses, her OS is improved. However, for a reader, the most interesting question would be, what might be factors that predict poor or good response. This has not actually examined here, although the authors say they have performed multivariate analysis on certain factors. The results of the cluster analysis, that the authors have performed, only shows that in each of these groups, there is a difference in OS between responders and non-responders, but this is I guess quite logical?
Instead (or in addition to) of a cluster analysis, I suggest doing a multivariate analysis on these 57 patients with meaningful variables, some of which would be:
- amount of residual disease after surgery (R2 if no surgery is performed)
- carcinosis/ascites/pleural effusion at diagnosis
- PFS interval under/over 6 months
As for treatments, the guidelines suggest using platinum combo in first line platinum sensitive disease (and this has been the practice here) but based on AURELIA study, the most effective treatment for platinum.refractory disease in weekly paclitaxel with bevacizumab, Why was this not used here? In particular, as this has a good effect on ascites formation. If this treatment was not used here, it should be mentioned as a weakness of this retrospective material.
Reviewer 3 Report
Comments and Suggestions for Authors
The article is well-organized, providing a clear structure with sections such as Background and Introduction, Materials and Methods, Results, and Conclusion. The abstract summarizes the study's key elements effectively. The research is well described, and the literature is properly cited.
Minor objections:
Material and methods: the authors stated: "Patients were diagnosed (high grades serous carcinoma and endometrioid) between 2016 and...(lines 133 and 134). However, in the Results, in Table 1. Study population baseline characteristics high-grade serous carcinoma are represented in 100% of cases and endometrioid 0%. This is unclear and confusing. If all 57 patients were diagnosed as high-grade serous carcinoma, why endometrioid carcinoma is mentioned at all?
In my opinion, in such a case, it would be better to use the term ovarian high-grade serous carcinoma than ovarian cancer. Ovarian cancer is a group of different diseases, with different outcomes and different therapeutic approaches.
